# Climate Change Detection and Attribution using observed and simulated Tree-Ring Width

Jörg Franke[1,2,*], Michael N. Evans[2,3,*], Andrew Schurer[4], Gabriele C. Hegerl[4]

[1]Institute of Geography, University of Bern, Switzerland.
5 [2]Oeschger Centre for Climate Change Research, University of Bern, Switzerland.
[3]Department of Geology and Earth System Science Interdisciplinary Center, University of Maryland, College Park, Maryland 20742, USA
[4]School of GeoSciences, The University of Edinburgh, Edinburgh, United Kingdom
* Both authors contributed equally to this study

10 *Corresponding author(s): Jörg Franke (franke@giub.unibe.ch) and*
*Michael N. Evans (mnevans@umd.edu)*

**Abstract.** The detection and attribution (D&A) of paleoclimatic change to external radiative forcing relies on regression of statistical reconstructions on simulations. However, this procedure may be biased 15 by assumptions of stationarity and univariate linear response of the underlying paleoclimatic observations. Here we perform a D&A study, modeling paleoclimate data observations as a function of paleoclimatic data simulations. Specifically, we model tree-ring width (TRW) observations as a linear function of TRW simulations which are themselves forward modeled from realistic singly-forced and cumulatively forced climate simulations for the period 1401-2000. Temperature and moisture-sensitive TRW 20 simulations detect distinct patterns in time and space. Temperature-sensitive TRW observations and simulations are significantly correlated for northern hemisphere averages, and their variation is attributed to volcanic forcing. In decadally smoothed temporal fingerprints, we find the observed responses to be significantly larger and/or more persistent than the simulated responses. The pattern of simulated TRW of moisture-limited trees is consistent with the observed anomalies in the two years following major 25 volcanic eruptions. We can for the first time attribute this spatiotemporal fingerprint in moisture limited tree-ring records to volcanic forcing. These results suggest that use of nonlinear and multivariate proxy system models in paleoclimatic detection and attribution studies may permit more realistic, spatially resolved and multivariate fingerprint detection studies, and evaluation of the climate sensitivity to external radiative forcing, than has previously been possible.

## 1 Introduction

One of the crucial questions in climate change research is to determine how external radiative forcings bring about climate variation and change, and if the forced response may be distinguished from the internal, unforced variability. To address this question, so-called "detection and attribution" (D&A) methods have been developed (Hegerl and Zwiers, 2011). Gillett et al. (2021). Generally speaking, D&A 35 studies match observed changes with patterns derived from climate model simulations driven by single and multiple external forcings, including solar variability, volcanic aerosols, the well-mixed greenhouse gases, orbital variations, and land use change. The idea initiated in early work by Hasselmann (1979). After methodological refinements and advances in climate modeling in the early 1990s (e.g. Hasselmann, 1993; Santer et al., 1993) there was growing evidence that the external greenhouse gas signal may be 40 differentiated from climate variability generated within Earth's climate system (Hegerl et al., 1996). Detection and attribution studies have been an important part of the Assessment Reports of Working Group I of the Intergovernmental Panel on Climate Change, from the calling for better detection of the role of human activities in climate forcing in the first Assessment report (1990), to formal detection and attribution studies comparing observed and simulated climate change in all Assessment reports since, with 45 increasingly confident assessments of the detection of human influences and estimates of the human contribution derived from attribution results.

Typically, D&A analyses have been limited to periods when instrumental observations of physically measurable variables and derived diagnostics are available, with global observation networks becoming dense enough for such studies about 100 to 150 years before present. This period allowed for attribution 50 of trends in many thermodynamic and dynamic characteristics of the climate system, including global and regional temperature, temperature extremes, ocean heat content, tropopause height, specific humidity, zonal mean precipitation, air pressure fields to potential forcings (e.g. Hegerl et al., 1996; Santer, 2003; Polson et al., 2013a; Bindoff et al., 2014; Eyring et al., 2021; Gillett et al., 2021). While 19[th] and 20[th] century instrumental observations cover a major increase in greenhouse gases and other human influences, studying the climate system response to non-anthropogenic external radiative forcings, such as

solar variability or volcanic eruptions, benefits from studying longer periods over which more realizations and/or longer-term processes are evident, and where the anthropogenic influence is less dominant. For instance, very few climatically important volcanic eruptions occurred in the past 150 years, but more than a dozen occurred over the past 600 years (Sigl et al., 2015) at nonuniform frequency in time, possibly creating long-term forcing of the climate system (McGregor et al., 2015; PAGES 2k Consortium, 2019; Brönnimann et al., 2019). Such longer-term studies would integrate longer-term responses of the climate system to external radiative forcing, enabling a more complete picture of the equilibrium and transient response, and ultimately of the climate sensitivity to external radiative forcing.

Paleoclimatology allows extension of the observational record into the past using indirect measurements of climatic conditions, which can be used to reconstruct past climate. Previous studies have detected a role of external forcing in the climate of the last millennium using annual mean surface temperature anomaly reconstructions on both a hemispheric scale (Hegerl et al., 2003; Schurer et al., 2013, 2014) and regionally (PAGES 2k-PMIP3 group, 2015). These analyses have found that volcanic forcing is detected with a smaller contribution from greenhouse gases that is detectable by 1900, and a contribution from solar forcing that was not detectable against climate variability. However, the reconstruction process itself introduces additional assumptions into detection and attribution studies that arise from the nature of the reconstructions, but which may not be justified. Many of these are demonstrated in pseudoproxy experiments (Smerdon et al., 2011) and through study of the extensive network of tree-ring width observations. These include assumed univariate, normally distributed and linear response of the paleoclimatic indicators to the target reconstruction variable (Evans et al., 2014; Wang et al., 2014); stationarity of patterns of regional and global scale climate variability (Wilson et al., 2010); seasonal and spatial representation (St. George, 2014; Smerdon et al., 2011); and autoregression characteristics in observations and target variables (Cook et al., 1999). Limited adherence to assumptions in observations and statistical modeling has been found to introduce biases into reconstructed variables, even in large scale averages (PAGES2k Consortium, 2017) and may lead to the underestimation of errors in D&A studies that are necessary to separate the forced and unforced responses (Neukom et al., 2019). In particular, autocorrelation due to memory in TRW affects the response to volcanism which, if not accounted for, biases D&A results (Lücke et al., 2019).

Progress in process understanding of paleoclimatic observations has led to the development of proxy system models (Evans et al., 2013), which may be used to identify systematic uncertainties and evaluate the extent of biases introduced by the reconstruction process into the D&A problem. One recent example is the Vaganov–Shashkin Lite (VSL) sensor model, which simulates standardized tree ring width (TRW) chronology variations based on monthly mean temperature, precipitation, and latitude. These inputs are used to estimate nondimensional growth arising from temperature and soil moisture conditions ($G_T$, $G_M$) either of which may stoichiometrically limit growth at each monthly time step: a multivariate and non-linear mimic of the processes by which forests sense and filter climatic variability and imprint those results in observable tree ring width variations (Tolwinski-Ward et al., 2011a, b). VSL has been widely tested for parameter estimation and global applicability.

Here we leverage VSL, historical gridded climate data products (Harris et al., 2014), singly and multiply forced climate simulations for the period 1401 to 2000 C.E. (Schurer et al., 2013), and the nearly 3000 consistently detrended TRW observations (B14, Breitenmoser et al., 2014) to perform an extratropical northern hemisphere D&A exercise directly using observed and simulated TRW data (Fig. 1, Eq. 1):

$$\alpha = \beta_0 + \beta_1 \, \alpha \tag{1}$$

With $\alpha$ representing the paleoclimatic observations (TRW in this case), and $\alpha$ representing the sensor modeled TRW simulations, themselves employing as input the output of a realistically forced climate model. Coefficients $\beta_0$ and $\beta_1$ represent, respectively, the unforced and forced amplitudes of variability (for a more detailed introduction, see Section 2.4 below). This approach stands in contrast to prior studies, which perform the D&A analysis in the space of reconstructed northern hemisphere mean surface temperature at annual resolution (Schurer et al., 2013, 2014). It has the potential advantages of circumventing assumptions required in the reconstruction process, and exploiting the "several-to-one" mapping that might reinforce environmental signatures in TRW data, such as spatially and temporally correlated patterns of moisture and temperature variability that mimic drought indices (Cook et al., 1999, 2004, 2010; Meko et al., 1995). Conversely, we may also identify key uncertainties in the sensor modeling, and the potential for the several-to-one mapping to obfuscate the detection and attribution of a forced response in the TRW observations.

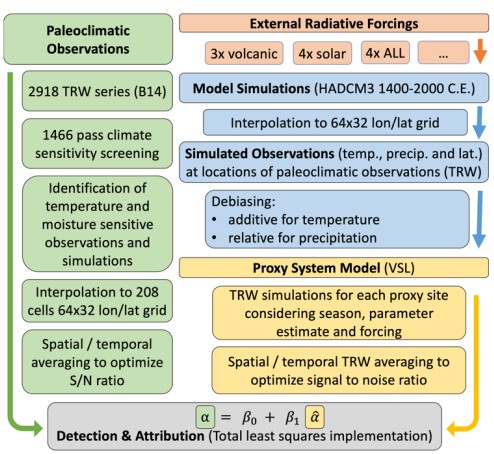

135

**Figure 1: Schematic overview of the performed analysis. General steps are indicated in bold, study-specific procedures in normal text. B14 refers to the Breitenmoser et al. (2014) data set, S/N stands for signal to noise ratio, T for temperature and PREC for precipitation.**

The remainder of this paper is organized as follows. First, we estimate and evaluate parameters for VSL, using gridded instrumental temperature (T) and precipitation (PREC) estimates and contemporaneous TRW observations (Section 2.2) in the period 1901-1970. Then VSL is used to build ensembles of simulated TRW series, in response to singly and cumulatively forced simulations of T and PREC, given uncertainty propagated through the parameter estimation process, and with bias corrections for simulated T and PREC ensembles (Section 2.3). We then estimate the D&A coefficients and their propagated uncertainty (Eq. 2; Section 2.4). The results are analyzed locally, regionally, and globally for detection and attribution of a forced climate response, vis a vis the simulated and actual TRW observations (Section 3). We discuss the results and the potential to extend the approach in Section 4; conclusions are drawn in Section 5.

## 2 Data and Methods

The inputs into and process of this detection and attribution study are illustrated in Fig. 1 and described below.

### 2.1 Tree-ring width measurements

We use the tree-ring width (TRW) collection described by and employed in (Breitenmoser et al., 2014), now referred to as B14, as the observational basis for the development and validation of VSL parameters, and as the D&A predictand (Eq 1). B14 consists of 2918 uniformly detrended and standardized tree-ring width chronologies from six continents and 163 species that have been upload to the International Tree-Ring Data Base (ITRDB, Zhao et al., 2019) until 2014. These series have been quality controlled for metadata errors, repetitive measurements, incorrect units, decimal point errors and misplaced positions (Tab. S1 in Breitenmoser et al., 2014). Detrending for biological age trends and stand dynamics and standardization to dimensionless growth indices was done in a hierarchical approach. If possible, negative exponential curves and linear regression curves of any slope were fitted. In case both methods failed, "a smoothing spline was fit with a 50% cut-off frequency at 75 % of each series length" (ARSTAN, Cook, 1985; Breitenmoser et al., 2014). Multiple measurements at the same site have been combined into robust means (Cook and Kairiukstis, 1990), which are variance adjusted for changing sample size through time (Osborn et al., 1997). For every point in time, which is explicitly resolved as one value per growing season each year, a chronology is based on at least 8 samples. We use the autoregressive-standardized (Osborn et al., 1997; Frank et al., 2007) version of the available chronologies from B14. We require that the chronologies have at least 40 years observed within the period 1901 – 1970 (see section 2.2 below).

We restrict subsequent analysis of simulations and the D&A exercise to the extratropical northern hemisphere continental areas, where the vast majority of TRW observations are located, with high concentrations in the North America, Europe and northern Asia (Fig. 2). Record length varies from 100-600

**Deleted:** is

**Deleted:**

**Deleted:** )

**Deleted:** .

**Deleted:** S

years (Fig. 2). Series availability is generally greatest between the mid-19th century and the late 20th century (Fig. 3), and the longest records are equally distributed in longitude across the north hemisphere boreal terrestrial latitudes (Fig. 2).

### 2.2 VSL parameter estimation

For the purpose of VSL parameter estimation, we use the global, gridded instrumental temperature and precipitation data sets CRU TS 3.23 (Harris et al., 2014), regridded to 64 longitude x 32 latitude (~5.6º) using a distance-weighted average of the four nearest neighbor values. To correct for mean temperature biases, we applied an adiabatic (-6 K/km) T correction to the regridded CRU product, based on differences between elevations of grid points and elevations of observed TRW chronologies (Evans et al., 2006). Parameters $T_1$, $T_2$, $M_1$ and $M_2$ describe the onset of growth (1) and point above which climate is no longer a limiting factor (2) for temperature (T) and moisture (M), respectively (Tolwinski-Ward et al., 2011a, 2013). We conditioned and validated all four parameters simultaneously using contemporaneous observations and VSL simulations within the period 1901-1970. The growth period is defined as a 16-month interval. To integrate monthly incremental growth arising from pre-season and growing season, the growth integration period starts in September of the previous year and ends in December of the current year in the northern hemisphere (previous March to current June for the southern hemisphere), the same period as in Tolwinski-Ward et al. (2011a) and Breitenmoser et al. (2014). Other VSL parameters are not calibrated, but taken from other studies (Evans et al., 2006; Fan, 2004; Huang et al., 1996; Tolwinski-Ward et al., 2011a, 2013; Vaganov et al., 2006; van den Dool, 2003). Within the chosen parameter estimation time window 1901-1970, with available N >=40, half of the years for which observed TRW data were available were chosen at random for parameter estimation ("calibration" Tolwinski-Ward et al., 2013). The other half were reserved for validation of the estimated parameters, via simulation using the estimated parameters, T and PREC not used to estimate the parameters, and comparison with the TRW observations withheld from the calibration process. The process was then repeated, using now the second half of data for parameter estimation (calibration), and the first half for validation of this parameter set. Up to 200 parameter sets were stored as valid, if all four calibration and validation correlations between simulated and observed TRW were all independently significant at the p<0.1 level; all others failing this validation test were discarded.

### 2.3 VSL simulations, 1401-2000

Temperature and precipitation input data for VSL are derived from climate model simulations. We use the set of simulations described in Schurer et al. (2014), which have been conducted with HADCM3 , interpolated to the same 64 x 32 grid as described in Section 2.2, to produce TRW simulations driven by singly and cumulatively forced climate simulations (Table 1). Because simulated T and PREC are spatially and seasonally biased relative to historical gridded T and PREC, we first bias-correct the HadCM3 T and PREC fields by computing T and PREC anomaly fields and adding them to (scaling them by) the CRU TS3.23 T climatology (PREC variability) for the overlapping period 1901-2000 C.E. This step also ensures that systematic differences in mean simulated T and PREC will not systematically bias VSL simulations based on parameter estimates conditioned on the historical CRU TS3.23 T and PREC products. Using the methodology as described in (Tolwinski-Ward et al., 2013), their Fig 8, we then identified the primary limiting factor for simulated growth (at p<0.05, assuming a binomial distribution) and divided the simulated chronologies into primarily temperature, moisture (M), both, or neither limited TRW simulations. The median, over parameter estimate realizations, of T and M-sensitive TRW simulations, were then separately weighted by inverse distance between observed and simulated grid point, observed expressed population signal (EPS, Wigley et al., 1984), and observed mean correlation between increment series within a chronology (RBAR, Cook and Kairiukstis, 1990) and averaged. Observed TRW were gridded and averaged in the same way as described above for subsequent D&A analysis (see Fig. 1 for a schematic overview of the entire process chain). Because centennial-scale climate variability may not be consistently preserved in the TRW records (Cook et al., 1995; Franke et al., 2013), and these timescales are poorly sampled in the 600 year period available for study, we removed low frequency variability by applying a 71-year high pass LOESS filter to both observed and simulated gridded TRW and focus our analyses on this residual interannual to multidecadal variance at annual resolution. The choice of annual resolution reflects the observational resolution (Section 2.1), the time-integrating nature of the sensor model (Section 2.2), the uncertainty of implementing radiative forcing estimates in the climate simulations (Gao et al., 2008), and the response timescale of the climate system to the large scale forcing. We call the results, on which we base the detection and attribution analyses, climate-sensor simulations. This nomenclature reflects modeling of both the climate in response to external radiative forcing(s), and the tree ring width observation that is basis for the comparison with actual TRW observations.

**Table 1: All forcing and single forcing HADCM3 simulations as well as control runs used in this study (V: volcanic, S: solar, G: greenhouse gases, L: land-use, A: tropospheric aerosols).**

| Number of simulations | Forcings | Period |
|---|---|---|
| 2 | NO forcing | 1401– 2000 |
| 4 | V, S, G, L, A | 1401 – 2000 |
| 3 | V | 1401 – 2000 |
| 4 | S | 1401 – 2000 |

240

### 2.4 Detection and Attribution

To solve for the D&A coefficients in (Eq. 1), we use the total least squares (TLS) D&A technique to account for errors in both dependent and independent variables within the regression procedure (Allen and Stott, 2003) to account for internal variability in both observations and model simulations. We follow the analysis used in Allen and Stott (2003), Polson et al., (2013a), and Schurer et al. (2013), which estimates a best fit regression coefficient (β) given by the equation:

$$\alpha = \beta \, (\alpha - \nu) + \nu \qquad (2)$$

In this study, $\alpha$ are the simulated tree-ring widths and $\alpha$ are the observed tree-ring widths, either at each grid box or spatially and/or temporally aggregated to increase the signal to noise ratio. $\nu$ are realizations of internal variability. Confidence intervals are obtained with the bootstrap method described in DelSole et al. (2019). They are calculated by randomly sampling, with replacement, pairs of values from the arrays of observed and simulated tree-ring widths to form new arrays the same length as the originals. A new scaling factor is then calculated by regressing the resampled model onto the resampled observations to represent uncertainty due to random noise. This process is repeated 10000 times and a 5%–95% confidence interval is estimated from the distribution. If the distribution of beta values is significantly greater than 0 (p<0.05) then the effect of the response to the forcing is considered to have been detected. If the distribution of β-values is significantly less than unity, the response in the climate-sensor simulations is too large; the response in climate-sensor simulations is significantly greater than observed, and the simulated climate sensitivity is smaller than observed. Conversely, if scaling range is significantly greater than unity, the simulated climate-sensor response is significantly smaller than observed in TRW, and the climate sensitivity of the model may be inferred to be larger than observed.  The estimate of the unforced variability as the residual of the D&A regression model provides another important result that needs to be compared with unforced variability of climate simulations (control runs) as a check of variability (PAGES 2k Consortium, 2019).

265

## 3 Results

### 3.1 Parameter estimation, TRW simulations and TRW observations

1664 of 2761 TRW chronologies in the B14 compilation were climate sensitive and therefore successfully simulated and retained for further analysis. With small differences between climate simulations, we found that 21% of the successfully simulated chronologies are temperature sensitive, ca. 57% are moisture sensitive, ca. 11% are both moisture and temperature sensitive, and ca. 11% are not climate sensitive, i.e. neither moisture nor temperature sensitive (Fig 2).  Distributions of temperature, moisture, both temperature and moisture, and neither temperature nor moisture sensitivity overlap in space. There are many moisture-sensitive TRW chronologies found in North America, the Mediterranean and other arid regions (Fig. 2, top right panel). However, there are also temperature-sensitive chronologies (upper left panel) and mixed responders (lower left panel) which are collocated in arid regions (upper left panel). Chronologies found to be neither temperature nor moisture sensitive (Fig 2, lower right panel) tend to be found at the highest latitudes, but not exclusively so.

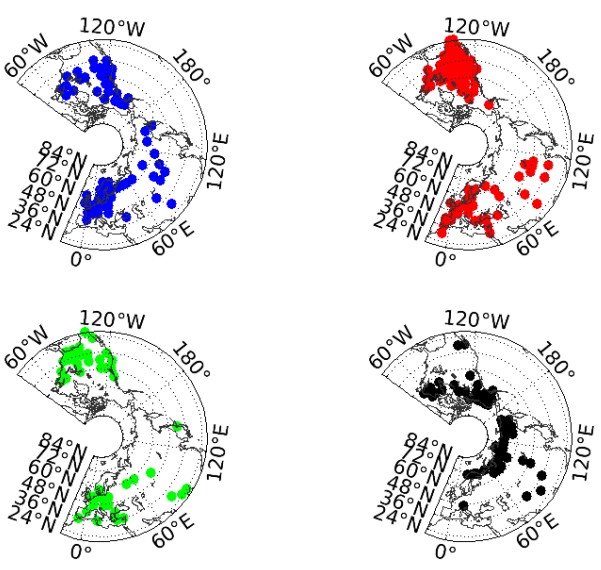

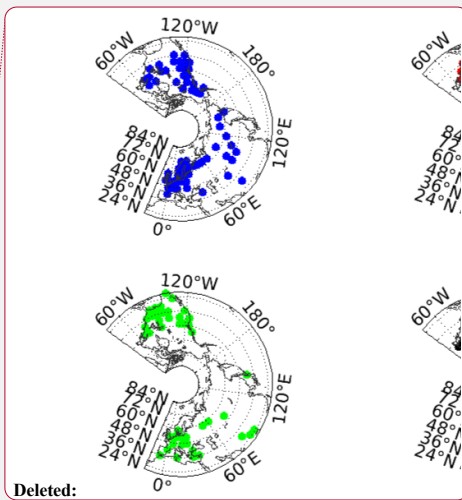

**Deleted:**

**Figure 2: Limitations determined for all TRW chronologies with valid parameter sets, separated into temperature sensitive (top left), moisture sensitive (top right), both T and M sensitive (bottom left) and neither T nor M sensitive (bottom right). (Maps created with Matlab mapping package M_map (Pawlowicz, 2022))**

We found that bootstrapped VSL parameter estimates were in many cases distinctly non-normal in distribution for some or all of the four parameters, and for some TRW simulations. Distributions were sometimes uniformly distributed across the prior expected parameter ranges, unimodal non-normal, and even bimodal. Because there were not necessarily well-defined means or medians across parameter sets and simulations, we used all valid parameter sets to produce TRW simulations. Hence, we propagate uncertainty arising from stochastic variation in the climate simulations through parameter and structural uncertainty in the ring width sensor model.

Because the fingerprint of external radiative forcing may or may not be distinct and unique in temperature and moisture, we use the fit of VSL diagnostic variables $G_T$ and $G_M$ (estimate of nondimensional growth arising from temperature and soil moisture conditions, respectively) to binomial distributions to determine whether each simulation is primarily controlled by temperature, moisture, both or neither control at the $p < 0.05$ significance level (Tolwinski-Ward et al., 2013). We perform a similar analysis to determine the same primary growth controls in the TRW observations, using the same diagnostics from the parameter estimation exercise. We then average TRW observations to the simulation grid resolution for temperature and moisture-limited simulations separately. Where there are multiple observed TRW chronologies available within a particular grid box, we construct a weighted average using inverse grid-point distance and intra-chronology mean incremental growth series correlation as weighting factors.

TRW simulations (Sections 2.2, 2.3) are developed for all locations, where TRW observations exist and the parameter estimation has been successful, i.e. for most of the extratropical northern hemisphere (Fig. 3). We exclude the southern hemisphere in this because only 5 temperature-sensitive chronologies and 1 moisture-sensitive chronology are located there. The record length of the simulations is constrained by TRW observations (see 2.1 and Fig. 3). The longest records are equally distributed in longitude across the north hemisphere boreal terrestrial latitudes (Fig. 3). Thus, statistics assessed across the simulations and observations are best described as representing the northern hemisphere temperate and subpolar terrestrial regions.

Furthermore, we note that the locations of temperature and moisture-sensitive chronologies overlap (Fig 2) but are generally not coincident (Fig. 3): for either the observed or simulated sets, only about 1/3 are both coincident in space and significantly correlated with each other (Fig. 2). Hence, for the remainder of the analysis presented here, we develop and discuss the temperature and moisture-sensitive results separately.

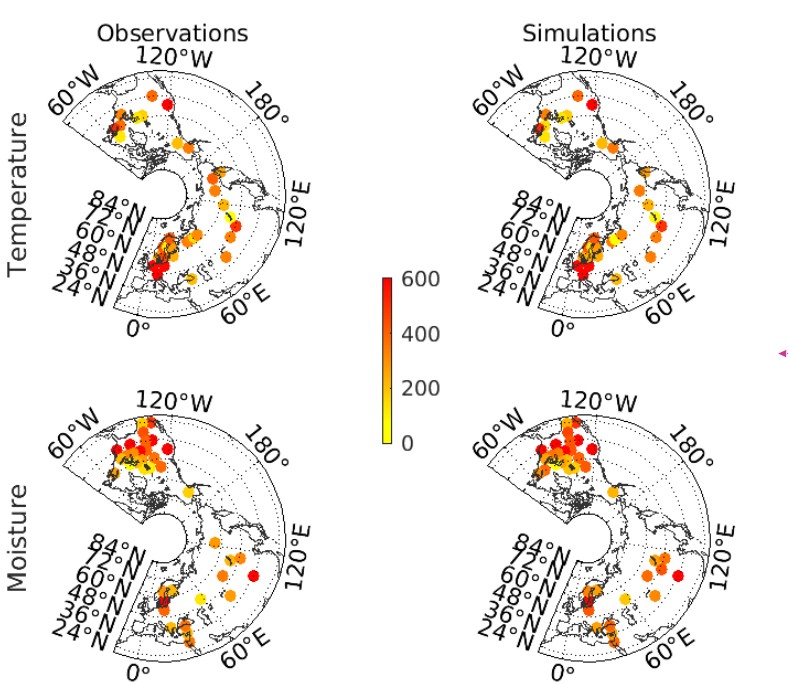

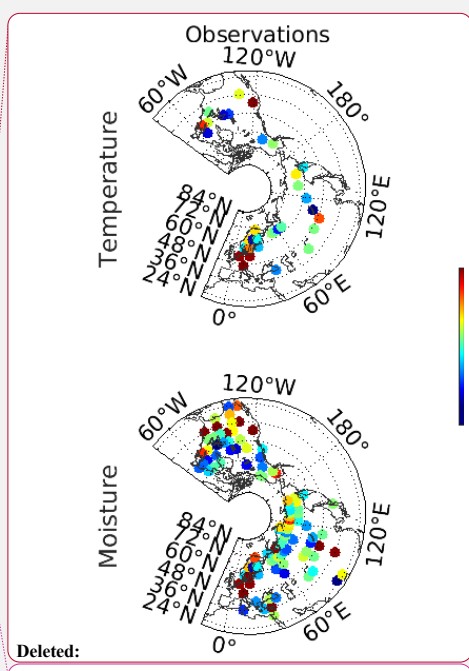

**Figure 3: Numbers of years (color scale) available for comparison between gridded, observed and climate-sensor simulated TRW chronologies, within the period 1401-2000. Top row: observed (left) and simulated (right) temperature-sensitive chronologies. Simulated chronologies are masked by observational availability, and in some cases (e.g., subpolar Eurasia) are neither moisture nor temperature-sensitive or else are both temperature and moisture sensitive, and do not appear in either simulation map. Lower panels: as for top panels, except for moisture-sensitive chronologies. (Maps created with Matlab mapping package M_map (Pawlowicz, 2022))**

Figure 4 shows the growth functions GT and GM for the subsets of temperature and moisture-sensitive identified TRW chronologies (section 3.1). Although VSL has well-known limitations, for instance the lack of a soil moisture model allowing for snow, and despite the potential for an unrealistic and coarsely resolved annual cycle in the HadCM3 simulations, the results suggest plausible seasonality of the growth response of the TRW simulations. In particular, GT for T sensitive chronologies is maximum but limiting in June-October with a median response (black line) maximum for July-September. GM for the T sensitive subset of chronologies is not limiting through the same period. Similarly, for M sensitive chronologies, GM is limiting between July-December. GE (the scaling associated with insolation (energy) as a function of latitude) is limiting (GE <0.7) after September for latitudes poleward of 20N (results not shown), and GT is not limiting through the warm months.

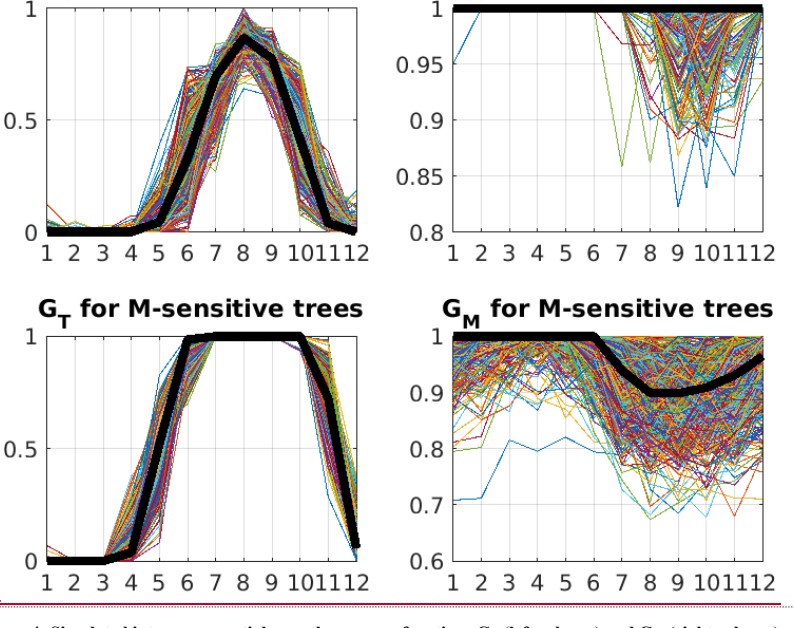

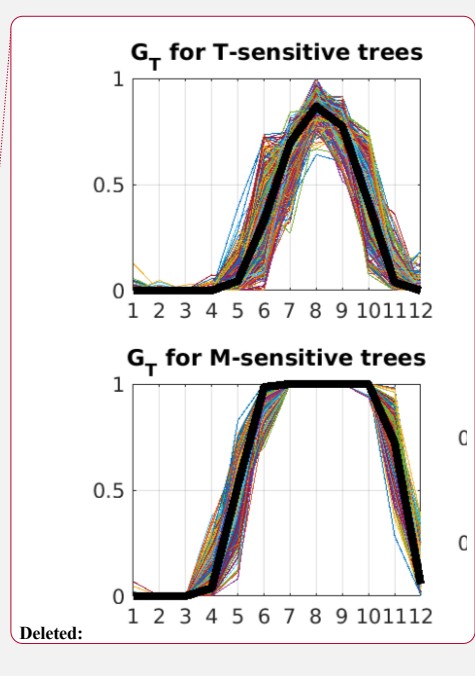

**Figure 4: Simulated intra-year partial growth response functions G_T (left column) and G_M (right column) for T sensitive (top row) and M sensitive (bottom row) simulations using ALL climate simulations, with parameters conditioned and validated using observed TRW data within the period 1901-1970. Heavy solid lines are median values across all simulations.**

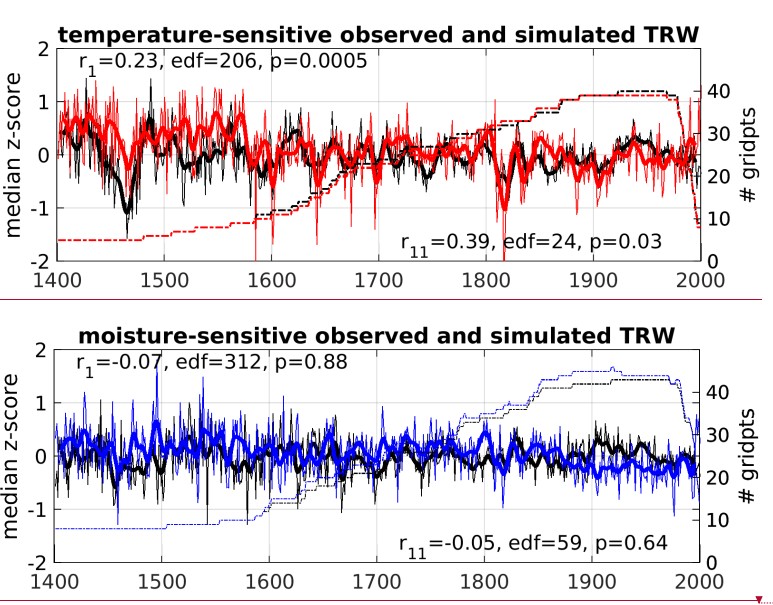

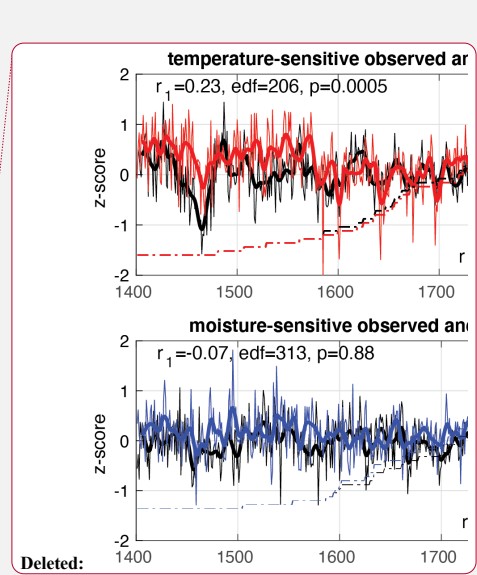

**Figure 5. Top: mean series for observed (black) and ALL-forcing simulated temperature-sensitive chronologies (red). Annually resolved and a 11-year Hanning window filtered time series are shown with thin and bold lines, respectively. Labels quantify the Pearson correlation (r), effective degrees of freedom (edf, Hu et al., 2017) and the p-value. x-axis units are years CE. Bottom: the same above but for moisture-sensitive observations (black) and simulations (blue). Note, this plot shows the global means of standardized TRW at all grid boxes with data before high-pass filtering and variance adjustment. There are small differences in numbers of observed and simulated chronologies that arise from both the observational masking and from the simulation parameter validation procedure (Section 2).**

**3.2. Comparison of observed and sensor-simulated TRW**

To detect an external-forcing signal in noisy, local observations, the signal-to-noise ratio has to be enhanced. This is commonly achieved by averaging in space and/or time (Sections 1, 2.4). We begin with analysis of global mean TRW variability at all locations where tree growth is either temperature or moisture limited, for comparisons between TRW observations and climate-sensor simulations driven with ALL forcings (Table 1). The variance of the average over all grid boxes increases back in time because of the decreasing numbers of records (Fig. 5), likely reflecting increasing uncertainty; the variance in the beginning of the 15$^{th}$ century is twice as large as that observed at the end of the 20$^{th}$ century. To reduce sensitivity of the detection and attribution analysis to observational uncertainty, we homogenize the variance through time by multiplication of a time dependent factor that is estimated by linear regression of the observed variance on the variance of TRW climate-sensor simulations from the control simulations. Results suggest limited but significant correlation between global mean observed and simulated TRW temperature-sensitive simulations, for both annual and decadally-filtered series (Fig. 5). Nonsignificant correlations are found for moisture-sensitive observations vs simulations at both annual and decadal timescales (Fig. 5). We find similar results for correlations between VOLC-forced simulations and temperature and moisture-sensitive observations (T sensitive: $r_1$=0.22, edf=201, p=0.001; $r_{11}$=0.48, edf=19, p=0.02; M sensitive: $r_1$=0.01, edf=380, p=0.40; $r_{11}$=0.00, edf=54, p=0.51). Correlations are not significant for comparisons between observed and SOLAR-forced or unforced TRW simulations (results not shown).

Based on these results, we test for detection of patterns in the TRW following volcanic eruptions in temperature and moisture-sensitive TRW chronologies, using a composite analysis across the 7 largest (above 95th quantile) volcanic forcing responses for events between 1670 and 1970 (Fig. 6). We show the composite for observations based on two forcings, stratospheric Atmospheric Optical Depth (AOD) reconstructed by Crowley and Unterman (2013) that was used to force the climate simulations (Fig 1; Table 2) and the more recent and probably more realistic inferred Global Volcanic aerosol Forcing (GVF, in W/m$^2$) by Sigl et al. (2015). We do not use the full period back to 1401 because only a few locations have data reaching that far back in time. However, including up to 12 eruptions where available leads to a very similar pattern (not shown). Consistent with the results for observed averaged temperature-sensitive chronologies, we find a reduction in simulated tree growth in the first two years after the eruption in nearly all locations worldwide (Fig. 6, top right), with perhaps a more statistically significant cooling response in the Crowley and Unterman (2013) event chronology. Observed tree growth at the temperature sensitive sites is reduced in most locations, but not as homogenously as in the simulations (Fig. 6, top left). Possible reasons may be related to the small sample size, uncertainties in the reconstruction of the volcanic forcing (Sigl et al., 2015), a low climate signal-to-noise ratio in ring width, and an enhanced signal-to-noise ratio in the simulations, which are represented by their 3-member ensemble mean (Table 1). Additionally, moisture influences may not be perfectly removed from the temperature-sensitive observations, because some of the positive growth anomalies appear in locations for which tree growth tends to be generally moisture-limited (in southwestern North America, northern European lowlands and the eastern Mediterranean (St. George and Ault, 2014). Thus, the composite observed temperature-sensitive response may in part also reflect increased moisture in dry regions following volcanic eruptions (Iles and Hegerl, 2015).

**Table 2: Largest volcanic eruptions used in the composite analysis based on atmospheric optical depth.**

| Crowley and Unterman, 2013 | | | | | | | | | | | |
|---|---|---|---|---|---|---|---|---|---|---|---|
| 1442 | 1456 | 1594 | 1600 | 1641 | 1673 | 1694 | 1809 | 1815 | 1832 | 1884 | 1903 |
| Sigl et al., 2015 | | | | | | | | | | | |
| 1453 | 1458 | 1595 | 1601 | 1641 | 1695 | 1783 | 1809 | 1815 | 1832 | 1836 | 1884 |

For our moisture-sensitive comparison, we do not find a global volcanic response of the same sign, but rather regions with uniform responses (Fig. 6, bottom row). The simulated Crowley and Unterman (2013) based event composite (Fig 6, bottom row, rightmost panel) produces positive growth anomalies around the Mediterranean and in western North America, and negative anomalies in Eurasia and eastern North

America, with more prominent composite positive regions than negative regions. The observed composite based on the Toohey and Sigl (2017) chronology produces no negative composite response in eastern North America and a small positive response in southwestern North America, the latter of which is consistent with simulations (Fig. 6, bottom row, middle and right panels).

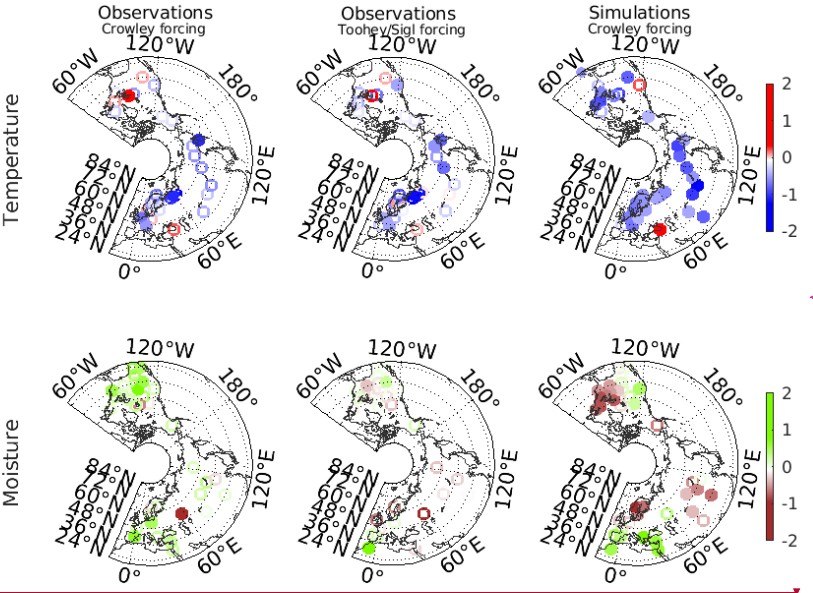

**Figure 6: Composite average ring width anomaly (standardized units) in temperature-sensitive TRW chronologies in the first two years after volcanic eruptions in observations and VOLC-forcing simulations (top). Because relatively few TRW records are available for 1400-1700 (Fig 2), the composite includes the 7 strongest eruptions between 1670 and 1970 based on the eruption chronologies of Crowley et al. (2013) (left and right column) and Sigl et al. (2015 middle column). Bottom row: as for top row, except for moisture-sensitive TRW observations and simulations. Closed circles indicate statistical significance (p-values of t-test < 0.05)**

### 3.3 Detection and Attribution analysis

We detect and attribute a response to volcanic forcing in both, the spatial mean temperature timeseries and the spatio-temporal pattern of moisture limited tree-ring records. As large volcanic eruptions disturb the climate system for a few years, we show results 3-year and 11-year moving averages. For the 3-year smoothed temperature-sensitive TRW averaged over all grid boxes, we find a significantly detectable scaling factor $\beta$ not significantly different from one; in other words, observed and simulated temperature sensitive chronologies agree within uncertainty (Fig. 7, left panel). This is true for both, the all-forcing and the volcanic-forcing based TRW simulations ($\beta_{ALL}$ and $\beta_{VOLC}$). For decadal averages (Fig. 7, middle panel), both scaling factors are likely greater than 1, and indicate that the observed responses are larger and/or more persistent than the simulated responses. Increased persistence is consistent with superposed epoch analysis results for volcanic eruptions using treering data (see Lücke et al., 2019).

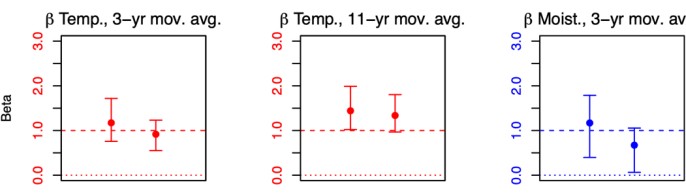

**Figure 7: Left: Beta values and uncertainties for 3-year moving averages (following DelSole et al., 2019) in the TLS D&A analysis for temperature-sensitive TRW (Fig 2, top panels). ALL and VOLC indicate regression on the all-forcing and volcanic-only based TRW simulations, respectively. Middle: as in left panel, except for 11-year running means; uncertainties adjusted for serial autocorrelation. Right: as in left panel, except for moisture-sensitive TRW (Fig 2, bottom panels) but with the aggregate mean response grouped by the two regions based on the positive or negative response of the TRW simulations (Fig 6, bottom right). Uncertainty ranges are based on 90% confidence intervals of scaling factors.**

As described above, moisture sensitive trees show positive growth anomalies in some regions and negative anomalies in other regions. We define a two-region spatio-temporal pattern identified in the moisture sensitive TRW simulations (Fig. 6, bottom right). Again scaling factors not significantly different from one for both ALL and VOLC forced simulations (Fig. 7, right panel) allow us to attribute moisture changes in response to volcanism.

**4 Discussion**

Detection and attribution studies using paleoclimatic data have previously focused on regression of reconstructed climate variables on realistically forced climate simulations (PAGES 2k Consortium, 2019; Schurer et al., 2014). In this study, we have attached a validated, realistically multivariate and nonlinear,
intermediate complexity proxy-sensor model (Evans et al., 2013; Tolwinski-Ward et al., 2013, 2015) to enable the D&A framework within the space of the paleoclimatic observation – in this study, tree ring width chronologies. Because this particular sensor model is a scaled and time-integrated transformation of temperature and precipitation variations into a single diagnostic, which is commonly observed across the terrestrial landscape, the potential for fingerprinting either distinct univariate or integrated plant-
stress-like signatures of the different radiative forcings becomes possible. The approach also substitutes structural and parametric uncertainty in the sensor model for the uncertainty arising from inversion of multivariate paleoclimatic observations for univariate climatic reconstruction, and so provides a complementary assessment of the uncertainty that propagates into the D&A results.

We find that the global mean forced response in temperature-sensitive TRW chronologies is consistent
with observations within the 1401-2000 period, a result that supports the prior work using global mean surface temperature reconstructions as predictand (Hegerl et al., 2006 and references therein), and implicitly the use of temperature-sensitive TRW chronologies for producing those results. However, we also find that moisture and temperature sensitive chronologies (Fig. 3) form distinct subgroups in space (Fig. 2) and in temporal average (Fig. 4). The fingerprint of climate forcing, as determined by comparison
between all-series averaged temperature and moisture sensitive observations and simulations is statistically significant in temperature, but not in moisture, for both ALL and VOLC forced simulations (Fig. 7).

For the attribution analysis targeting volcanic forcing (Figs. 6, 7), we find disagreement in the amplitude of the temperature-sensitive forcing as a function of timescale, with the observed annual/3-year/decadal timescale variance being smaller/equal/greater than the simulated variance (Fig. 7, left and middle pan-
els). One explanation would be that the simulated peak temperature response to volcanic forcing is unrealistically large. This has been observed for the HadCM3 climate model simulation in a previous study (Schurer et al., 2013). Volcanic forcings used to produce the climate simulations may also be oversimplified in time and/or space relative to actual forcing (Stevenson et al., 2017), or its timing may be incorrect (yielding suppressed amplitude in reconstructions). For many eruptions with an unknown date, the
eruption was set to January 1 and the AOD is entered into the model in four equal latitude bands only, proportional to the amount of Sulphur in the Antarctic and Greenland ice cores (Crowley and Unterman, 2013). Because TRW simulations are a simplified representation of actual TRW variation, they neglect the observational uncertainty and the potential for superimposed and competing influences, such that the
simulated TRW response to forcing may be relatively large. This is indeed the case; for either VOLC or ALL forcing, simulated variance is about one-third larger than observed variance (results not shown).

**Deleted:** values

**Formatted:** Font: Times New Roman, English (UK)

A further explanation could be that autocorrelations in observed and simulated TRW are different. We find observed mean TRW autocorrelation to be about two-thirds larger than that of VOLC forced simulations (results not shown). Consequently, we find the observed TRW variance at decadal resolution to be significantly greater than simulated TRW variance. This result suggests that i) the observed response contains decadal timescale non-climatic variation not adequately removed by observational signal processing (Cook and Kairiukstis, 1990), ii) mechanisms represented in the climate simulations are inadequate to represent slower response timescales of volcanic forcing (Miller et al., 2012), iii) mechanisms of forest response to volcanic forcing via soil moisture, air temperature or insolation variations, as represented in VSL, are insufficient to represent the observed lower frequency response (Esper et al., 2015; Lücke et al., 2019) or a combination of all three factors. Previous studies found scaling factors to increase as more smoothing is applied (Schurer et al., 2013). However, they did not reach the point of a significantly larger response in observations than simulations.

Previous studies based on historical observations found that volcanic eruptions produced positive precipitation and streamflow anomalies in the Mediterranean and the southwestern United States, whereas negative anomalies were observed at high latitudes, and in western North America, the Indian and southeast Asian region and the tropics (Iles et al., 2013; Iles and Hegerl, 2015). This is in agreement with the CMIP5 simulated precipitation response (Fig. 1a in Iles and Hegerl, 2015), although the pattern in observed precipitation was very noisy and not clearly observed. In contrast, the response was identifiable in observed streamflow data which covers a longer period and integrates the precipitation response. Reasons that the precipitation response couldn't be detected are likely to include the small number of eruptions in the instrumental period over which a composite was formed, combined with low signal-to-noise ratio for precipitation (Fig. 1a in Iles and Hegerl, 2015), and the complex precipitation response pattern with regions of increases and decreases which is more difficult to detect (see also Polson et al., 2013b). We would obtain similarly non-detection and non-attribution were we to define regions manually (e.g. Northern Europe vs Mediterranean or Western vs Eastern North America), or for a smaller integration over the years following an eruption. This finding is in agreement with (Rao et al., 2017), who see the effect in tree-ring reconstructed PDSI only in a very small region of north western Europe, southern Spain and northern Morocco, and with Fischer et al. (2007), who found increased precipitation in the Mediterranean and Scandinavia, and decreased precipitation in Northwestern and Central Europe following volcanic eruptions, although not statistically significant in many locations and accompanied by high uncertainty in the reconstructed precipitation response. For such small-scale regions, our TRW network is too sparse, our simulation grid too coarse and the time span of the TRW series is too limited to calculate robust composites. The present study respects some of these challenges by extending the analysis several centuries into the past (Table 1), integrating the forced response over time and space (Fig 7), and forming the attribution model using the native observed variable rather than a reconstructed climatic variable (Section 1; Fig. 1). We find a similar pattern in moisture sensitive TRW (Fig 6). Simulations are most consistent with the expected pattern if the composite is based on the same forcing chronology as that used to drive the underlying HadCM3 simulations (Crowley and Unterman, 2013). The pattern in TRW observations agrees better with the more recent volcanic forcing chronology of Sigl (2015). This suggests the latter forcing series reconstruction may be more consistent with the response as observed in TRW. However, the two forcing chronologies are similar enough, that the two-region detection and attribution analysis (Fig 7, right panel) produces the significant detection of both the ALL and VOLC forced TRW signals, within uncertainty of unity, lending support to the conclusions of (Iles and Hegerl, 2015).

**5 Conclusion**

We have estimated the contribution by all forcing and volcanic forcing to treering data, based on a detection and attribution study using observed and modeled tree-ring width data directly for the exercise. We found that temperature and moisture sensitive TRW contain different signatures of the forced climate response over the past six centuries. Specifically, we find that the signature of the ALL- and VOLC-forcing response is most evident across the mean of all temperature-sensitive chronologies, but not across the mean of all moisture sensitive chronologies. The amplitude of the temperature-sensitive forced response is larger than expected from the model simulations in decadally filtered results, suggesting inaccuracies in the representation of forcing and/or response on those timescales in observations, simulations, or both sources of information. Additionally, we detect and attribute a previously identified spatial pattern in moisture-sensitive response to volcanic forcing at annual timescales, with a dipole drying/moistening pattern similar to the one previously identified by others within the historical time period and with direct moisture observations. In this study we demonstrate for the first time that climate change D&A can be conducted directly on paleoclimatic observations and their multivariate, non-linear proxy system simulations, allowing for a much more reliable model evaluation than possible if using reconstructed

climate variables. The results may realistically diverge from those obtained by D&A studies using uni-
variate surface temperatures reconstructed from similar datasets, because the underlying observations
may in reality be multivariate, nonlinear responders. Further studies could improve upon this proof of
concept by incorporating stable isotopic observations in combination with isotope enabled climate model
simulations, and by accessing a longer time interval for developing composite analyses, additional data
types, and a larger ensemble of realistically forced climate simulations.

**Data availability**
The HADCM3 simulations are available at the Center for Environmental Data Analysis: https://cata-
logue.ceda.ac.uk/uuid/9b51eb5d44524e4c9aebcbbc53d79a27. Gridded instrumental data is available at
575 the Climate Research Unit: https://crudata.uea.ac.uk/cru/data/hrg/. The B14 TRW data collection is
available at the World Data Center for Climate (WDCC) at DKRZ, where it was used as input data for
a data assimilation based climate reconstruction called EKF400 (Franke et al., 2017): http://cera-
www.dkrz.de/WDCC/ui/Compact.jsp?acronym=EKF400_Input_Data_v1.1. Results illustrated in Fig-
ures 2-7 are available from the NCEI/WDS for Paleoclimatology, landing page:
https://www.ncei.noaa.gov/access/paleo-search/study/36773

**Author contribution**
JF and MNE designed the study with contributions of GCH. JF prepared the inputs for the TRW simula-
tions, which were conducted by MNE. JF performed the D&A analysis with support of AS. JF and MNE
wrote the paper with help of AS and GCH.

**Competing interests**
The authors declare that they have no conflict of interest.

**Acknowledgements**
This study originated in a sabbatical stay of MNE at the University of Bern supported by the Oeschger
Center for Climate Change, the Sigrist Foundation and the University of Bern, Department of Geography.
JF was funded by the Swiss National Science Foundation (project 162668) and by the European Union
(H2020/ERC grant number 787574 PALAEO-RA). A.S. and G.H. were funded by the UK Natural En-
vironmental Research Council via the grant Vol-Clim (NE/S000887/1), GloSAT *(NE/S015698/1)* and
595 under the Belmont forum, Grant PacMedy (NE/P006752/1). We thank the British Atmospheric Data Cen-
tre (BADC) for access to the HadCM3 climate simulation (http://badc.nerc.ac.uk/browse/badc/euro-
clim500). MNE is especially grateful to Martin Grosjean, Raphael Neukom, Christoph Dätwyler, Jörg
Franke, Stefan Brönnimann and their research groups for stimulating conversations, chocolate, coffee
and emotional support through a particularly difficult life transition; and to Anupma Gupta (1971-2015),
and Aditi and Maya for their ongoing teachings and wisdom.

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
