# Peer review of "Climate Change Detection and Attribution using observed and simulated Tree-Ring Width"

_Climate of the Past, 2021_

## Author Comment (AC1)

We are very grateful for this thorough and constructive review.

In general, Reviewer 1 requested

- More background and citation of the detection and attribution literature.
- Clearer separation of methods from results.
- Expanded explanations of the approach: selection of proxy observations for the D&A exercise
- A clearer conclusion, not "only assumptions".

Our answers to the reviewer comments are highlighted in bold below.

General comments (Reviewer 1)

The paper addresses the relevant scientific questions related to the evaluation of model simulations and observed data, and within the scope of CP. Only one citation Hegerl and Zwiers, 2011 is not enough and even was discussed earlier in IPCC 2001 Detection of Climate Change and Attribution of Causes — IPCC. Reports TAR, Climate Change 2001, 2013 AR5 Climate Change: The Physical Science Basis Detection and Attribution of Climate Change: from Global to Regional.

**We will add further introduction to the development and evolution of D&A research, including a set of key references at the end of the first paragraph of the introduction. We also will note that detection and attribution of climate change has been discussed and summarized in every IPCC WG1 assessment report to date. An exhaustive review of such work is beyond the space available in the present manuscript, but is published elsewhere as cited (Hegerl and Zwiers, 2011).**

Some parts of the methodological approaches are mixed up with results. The authors often avoid clarifications and explanations, which can be helpful. There is no clear conclusion, only assumptions. It is not clear how TRW chronologies were pre-selected, which method of standardization was applied. Moreover, all chronologies have different age trends, periods, site-specific, and species-specific differences. All uncertainties can be related to the methodological approaches and pre-selection procedure.

**We will provide further clarifications and explanations in the revised manuscript, including: TRW chronology selection and most importantly, validation of the TRW simulations we perform. We also**

**refer the reader to Breitenmoser et al (2013), which addresses these questions directly, and on whose data compilation our study was based and builds upon.**

Specific comments

L. 29 Starting from the first sentence - brings confusion between different forcing and factors. The main research question/hypothesis in the article can be better formulated. It is not clear if the main focus is irradiative forcing before /after volcanic eruptions or in general forcing or any other forcing factors or mechanisms that will be taken into consideration. It should be clearly formulated.

**We regret not more clearly stating the goal of the study in the abstract.  We will revise the relevant sentences in the abstract to:**

**"Here we perform a D&A study, modeling paleoclimate data observations as a function of paleoclimatic data simulations. Specifically, we model tree ring width (TRW) observations as a linear function of on TRW simulations which are themselves forward modeled from realistic singly-forced and cumulatively forced climate simulations for the period 1401-2000."**

**This reflects the statistical model for the detection and attribution study (equation 1) as well as the specific properties of the problem, which are defined in Figure 1 and Section 2.**

L. 80 unclear which time period (past 600 years from xx to xx?). Which grid net was used (lat, alt)? TRW observations – common period? Which chronologies, citations, how many chronologies n=? Pre-selection high latitudes, mid-latitudes? Please provide citations and refer to Fig 1.

**We will clarify the period "1401 to 2000 C.E.", number of chronologies, the focus on extratropical northern hemisphere because of the availability of the particular target observational data set, and added citations to sources for all of these elements.  Additionally, Fig. 1 will be revised to provide additional information about the spatial grid we used to compare observations to simulations. All of these elements are determined by the results of Breitenmoser et al (2014), whose attributes we now provide in summary for the reader in Section 2.1.**

L.89 no citation ." to prior studies", which one? Please cite. …" of reconstructed surface temperature"? Summer temperature? Annual temperature? Please specify.

**We will revise the text to refer here to the studies of Schurer et al (2013, 2014) and refer to the use of northern Hemisphere annual mean surface temperature reconstructions in that work.  One advantage**

**of the approach we take to the D&A problem is that we do not explicitly require, nor assume, the particular season or climate variable which is most likely reflected in the TRW observations (section 1: "It has the potential advantages of circumventing assumptions required in the reconstruction process...") Additionally, we will clarify in section 2.1 that by the nature of the observational data type, the temporal resolution is one observation per growing season: ("For every point in time, which is explicitly resolved as one value per growing season each year,…").**

L.101 – historical temperature. Is it reconstructed temperature? If yes, please provide the period. If not, please clarify.

**We will clarify that we use gridded instrumental temperature and precipitation product CRU TS3.23, 1901-1970 period, for the development of VSL parameter estimates.**

Subsection 2.1 It will be good to provide more details about the TRW database used for analysis, e.g., time period, regions, species.

**We will add details about the used tree-ring width data set, including number of species, data from all continents and information about the detrending and standardization. For further information we also refer the reader to Breitenmoser et al (2014), and note that many of the reviewer's concerns were shared by those authors, who attempted to develop a homogeneous data set free of biases arising from the factors noted by the reviewer.**

Subsection 2.2 Please explain what T and M mean. "Parameters T1, T2, M1 and M2.

**We will revise the text to better introduce the tuned parameters, as follows:**

**"Parameters T1, T2, M1 and M2 describe the onset of growth (1) and point above which climate is no longer a limiting factor (2) for temperature (T) and moisture (M), respectively (Tolwinski-Ward et al., 2011a, 2013)."**

**The development and validation of this model is described in Tolwinski et al (2011a, 2013) with reference therein to the more complex Vaganov-Shashkin model (Vaganov et al 2006, 2011) from which VSL is derived.**

L. 160 please clarify why a 71-year high-pass LOESS filter was applied.

**Besides the previous explanation that centennial scale variability is not preserved in many records in the tree-ring data set, citing Franke et al 2013 in support, we will also note that the period of study is**

**relatively short for analysis of centennial timescale variations to be statistically significant. This need for additional process replicates is stated in the Introduction as a motivation for paleoclimatic D&A studies, and its value is illustrated in the results (e.g. Table 2 and Fig 4). Should future studies be able to access a longer time interval of both realistically forced climate simulations and paleoclimatic observations, this restriction might be usefully relaxed, and we now note this in the Conclusion.**

L. 194 this description should be provided earlier in Figure 1 legend

**This description is already inside Figure 1.**

L. 210-220 info about TRW chronologies, length, sites should appear earlier in section 2, subsection 2.1

**We will describe the TRW observation network in more detail in section 2.1.**

L. 262 it is unclear based on which criteria the 12 largest volcanic events were pre-selected (VEI?) and which one (names). Please clarify.

**We will clarify that we use the 12 largest (above 95th quantile) volcanic event between 1401 and 1970, following Crowley and Unterman (2013). There event size is measured in stratospheric Atmospheric Optical Depth (AOD). This forcing is the same which is used to force the HadCM3 model simulations used in this study (Schurer et al 2013). Additionally, we use the more recent inferred Global Volcanic aerosol Forcing (GVF, in W/m2) by Sigl et al (2015). We included a table with the years of the 12 strongest eruptions in both forcing data sets. We did not add the names of the volcanos because some are still unknown.**

Figure 5. It is unclear why annual temperature and annual precip. are considered? In legend VOLC – volcanic forcing, in Table 1 – V. Please select one abbreviation through the whole manuscript.

**The detection and attribution analysis in Figure 5 is between annually resolved TRW simulations and observations (see also equation 1 and section 2.4), not between annually averaged temperature and precipitation. Because the TRW modeling indicates that the observations may be distinguished as either temperature or moisture limited, however (section 3.1; Figs 2-4), and also because the radiative forcings are applied only at annual resolution in the underlying climate simulations (Schurer et al 2013), we perform the D&A exercise for these subsets of TRW observations and simulations at annual resolution, respectively. A map figure showing the distribution of these subgroups by TRW observational location will be added to the manuscript.**

Technical corrections

**All points without a reply below will be corrected exactly as suggested.**

L.16 Abstract: tree ring width replace with tree-ring widthL.42 – references are not in the correct order. Please correct.

L.43 instrumental period of observations, please specify the period. For many stations outside Europe the instrumental period of observations for precipitation is rather short (ca. 50 years).

**The previous sentence explains the broad range of climatic variables, which have been used in D&A studies. The density and time period covered by each of these variables differs. Explaining all details is not our concern here in the paleoclimatic context. We want to focus on the additional gains of using a period much longer than any instrumental record. Therefore, we decided to leave this sentence more general.**

Figure 1. Abbreviations should be clarified in the Figure legend. E.g., optimize S/N ratio. Please clarify numbers (B14)? Please check abbreviations and provided an explanation in scheme precipitation or precip.,) in the text L. 101 (PREC). Please be consistent.

L. 105 – Eq. 1 is not in Section 2.4. Firstly, it was mentioned p.2. It should be Eq. 2. Please correct the numbers.

L. 110 consider revision .. "is constructed is illustrated"

Polson et al, (2013), replace with Polson et al. (2013),

L. 176, 178 – please check (is/are)

L. 203 GT, GM – please clarify what is what.

Fig 3, x-axis please write Year (CE)

Fig. 3 in plot – edf and citation edf – please clarify

**edf stands for "effective degrees of freedom" and is already described in the figure caption, with citation of Hu et al (2017) for further details.**

L. 232 replace to "..a 11-year.."

L. 341 AOD – please clarify.

L. 342 please add a citation.

Citation: https://doi.org/10.5194/cp-2021-80-RC1

---

## Author Comment (AC2)

We are very grateful for this thorough and constructive review.

In general, Reviewer 2 (Anchukaitis) requested

- Results that show the pattern of skill for simulating the observed TRW data, in particular for moisture-sensitive chronologies, and
- in particular the environmental dependence that was successfully simulated, considering also seasonal dependencies.

Our answers to the reviewer comments are highlighted in bold below.

Reviewer 2 (Anchukaitis)

This is a thorough and very interesting study combining proxy systems modeling with the detection and attribution framework applied directly to tree-ring width proxy chronologies. I especially appreciate all the work that authors have invested in dealing with the many challenges of the data, model(s), and simulation output (these are considerable). In particular, the attention to evaluating VSL in the 'real world' before moving into the simulation and D&A framework, observations about the nature of parameter sets (e.g. information around Line 195 is really interesting to think about the implications and potential interpretations of this), attention to temperature and precipitation bias issues, and various other aspects. This will be a useful touchstone paper and I suspect also motivate further work, since tree-ring proxy systems models are both valuable but then again challenging to use in frameworks such as the one here because of model bias, parameter uncertainty, and often mixed or weak climate signals in large tree-ring datasets (particularly for temperature) compared to the deterministic climate signals that emerge from VSL. My major comments below are primarily around the ability of VSL to simulate the chronology set here and how this prop[a]gates into the differences between observed and simulated series and how this then impacts particularly the moisture-sensitive D&A:

1. Patterns of successful simulations (Line 194 and elsewhere): I think it would be desirable to get a better idea of where and for what chronologies the VSL simulations are successful - I get the sense from the manuscript and the Tolwinski-Ward papers show that (in general and not surprisingly), VSL will do better when the chronology in question has a strong climate signal itself (because VSL is driven by climate filtered through some possibly nonlinear simulated processes). In any case, it would be helpful to visualize the success of VSL here - where (which chronologies, that is) can VSL successfully simulate

and how many of these are moisture vs. temperature - my guess would be that the majority or plurality of the Breitenmoser chronologies are moisture-sensitive or mixed sensitivity based on e.g. St. George 2014 and the original a quick look at the Breitenmoser paper - so, does VSL do really well with more (% wise) T or M limited sites? Are mixed sites generally not as well simulated? Some of this is likely already part of the original Breitenmoser paper, but this is useful information when evaluating where the observations and simulation (e.g. Figure 4) agree or disagree and what might be the potential reasons behind this.

**We used the approach of Tolwinski-Ward et al (2013, Fig 8) to diagnose the climate sensitivity for each simulation as the variable for which, at the p=0.05 level of confidence, the limiting sensor variable was T, M, both or neither. The results for T and M sensitivity are shown in Fig R1 (revised Fig 2) at a coarse 64 x 32 resolution, weighted for distance and chronology statistics, and in Fig 3 as a spatially averaged timeseries (section 3.1). Because of this gridding and averaging, we gain a robust basis for comparison with climate-sensor simulations, but lose the richness of information encoded in the individual TRW observations. The reviewer's sense is correct: of the simulated chronologies, for the ALL forcing scenario (Figs 3, 4), and neglecting small differences arising from climate simulation ensemble member (n=4) differences, for the 1583 successfully simulated chronologies, 21% are temperature sensitive, 57% are moisture sensitive, 11% are both moisture and temperature sensitive, and 11% are neither moisture nor temperature sensitive. Fig R1 (revised Fig 2) shows that there are both T and M sensitive chronologies distributed throughout the northern Hemisphere continents, but as we noted, only about 1/3 of the separated T and M sensitive chronologies are coincident. We will provide the additional information from this paragraph in the revised manuscript in Section 3.1. Note also that Fig R1 (revised Fig 2) has been revised for clarity and to correct a plotting error that left many M sensitive chronologies unplotted.**

[Figure]

*Figure R1 (revised Fig 2). Temperature (left) and moisture (right) sensitive tree-ring width (TRW) chronologies, as determined by the method of Tolwinski-Ward et al (2013) with p<0.05 significance. Colorscale indicates number of years available for comparison of observed and simulated TRW.*

Particularly for moisture, there is the question of the seasonality of the climate response vs. the seasonality of tree growth. For instance, in western North America and the Mediterranean, winter/spring moisture will be important for growth, while in Northern Europe and other parts of North American, annual or summer moisture will control moisture-sensitive tree growth. The extent to which VSL can do this adequately would seem to be key to making the connection from climate forcing (e.g. volcanism) to local climate to tree growth with as much confidence as possible.

**Figure R2 shows the growth functions $G_T$ and $G_M$ for the subsets of temperature and moisture-sensitive identified TRW chronologies (section 3.1). Although VSL has well-known limitations, for instance the lack of a soil moisture model allowing for snow, and despite the potential for an unrealistic and coarsely resolved annual cycle in the HadCM3 simulations, the results suggest**

plausible seasonality of the growth response of the TRW simulations.  In particular, $G_T$ for T sensitive chronologies is maximum but limiting in June-October with a median response (black line) maximum for July-September. $G_M$ for the T sensitive subset of chronologies is not limiting through the same period.  Similarly, for M sensitive chronologies, $G_M$ is limiting between July-December. $G_E$ (the scaling associated with insolation (energy) as a function of latitude) is limiting (<0.7) after September for latitudes poleward of 20N (results not shown), and $G_T$ is not limiting through the warm months.

[Figure]

*Figure R2: Simulated intra-year partial growth response functions $G_T$ (left column) and $G_M$ (right column) for T sensitive (top row) and M sensitive (bottom row) simulations using ALL forcing climate simulations, with parameters conditioned and validated using observed TRW data within the period 1901-1970.*

I was also surprised (e.g. Figure 4) by the lack of chronologies further to the west (the Great Basin, Sierra, California, etc) - these are some of the most moisture sensitive sites in the world - why are they not represented here? Is this a VSL problem? A model simulation data/bias limitation?

**Fig R3 (revised Fig 4) has been revised to correct for plotting errors, but the reason that there are few locations plotted is that there are relatively few chronologies that fully cover the entire 1401-2000**

**period for study (see Fig R1 and Fig R4). For example, in California (lon 125W-114W), there is only one moisture sensitive gridpoint simulated, although it covers the full 600-year comparison interval.**

[Figure]

*Figure R3 (revised Fig 4): Composite average ring width anomaly (standardized units) in temperature-sensitive TRW chronologies in the first two years after volcanic eruptions in observations and VOLC-forcing simulations (top). Because relatively few TRW records are available for 1400-1700 (Fig 2), the composite includes the 7 strongest eruptions between 1650 and 1970 based on the eruption chronologies of Crowley et al. (2013) (left and right column) and Toohey and Sigl (2017) (middle column), respectively. However, not all TRW records cover the full period. Bottom row: as for top row, except for moisture-sensitive TRW observations and simulations.*

[Figure]

*Figure R4. Left: Availability of M sensitive observations for 1650, 1750, 1850, 1950, and at right, for T and M sensitive chronologies for 1450 and 1550. Colors only indicate the value of the simulated TRW at each point in time and in space.*

Figure R5 shows the limitations determined for all TRW chronologies for which we found valid parameter sets for T1, T2, M1, M2 (sections 2.2, 3.1). As the reviewer expects, there are many moisture sensitive TRW chronologies, as determined by the methodology of Tolwinski-Ward et al 2013, in North America, the Mediterranean and other arid regions (Fig R2, top right panel). However, there are also T sensitive chronologies (upper left panel) and mixed responders (lower left panel) which are collocated in arid regions (upper left panel) at the level of coarse gridding we use in the D&A analysis (64 x 32 global resolution).

[Figure]

***Figure R5: Limitations determined for all TRW chronologies with valid parameter sets, separated into temperature sensitive (top left), moisture sensitive (top right), complacent (bottom left) and neither T nor M sensitive (bottom right).***

2. Regarding Figure 4 and results shown there: Are all the locations shown in these maps really places where (1) VSL successfully simulates the chronology/ies at the location and, (2) where there is a true T or M limited site? I ask because I find myself surprised, for instance, to see apparently T sensitive sites in mid-latitude or arid North America and parts of the Mediterranean, and note in particular that several of these T-sensitive sites show increased growth post eruption, suggesting perhaps these are not simple temperature sensitive sites in the real world (observations)? Whereas the simulation shows (as expected) a growth reduction everywhere. I wonder if the difference in observations and simulations for T sensitive locations can be explained by the strength of the confidence that some of these are really temperature sensitive? Again, I look at North America and find myself wondering if many of those mid-continent sites are sufficiently temperature sensitive to be confident they can be compared to VSL limited by temperature alone. Or, put another way, VSL (driven by climate) will have a strong temperature-mediated growth response if the parameters and local climate make the simulation at that location temperature sensitive (and, this also leaving aside landscape-scale changes in sensitivity, e.g. differential tree growth response in the same grid point - Bunn et al. (2018). Spatiotemporal variability in the climate growth response of high elevation bristlecone pine in the White Mountains of California. Geophysical Research Letters, 45(24), 13-312.).

As well in Figure 4, there seems to be several important and interesting mismatches for moisture sensitive sites as well - for instance, for Crowley et al. eruptions (left and right columns) in North America the simulations show drying/reduced growth in the northeastern United States and a negligible response on the central and western part of the continent, while the observations show the opposite - e.g. a negligle signal in the eastern/northeastern part of the country, and a wet anomaly in the central/west.  The authors do note some of these features (Lines 280 to 286), but what stands out to me for the purpose of this manuscript is the differences between simulations and observations even when the same forcing dataset is used in North America in particular.  Perhaps though the most consistent signal is indeed the European dipole (wet/more growth in the Mediterranean, drier/reduced growth in Northern Europe) - this latter feature somewhat consistent with Fischer et al. 2007 (10.1029/2006GL027992) and more so I think with Rao et al. 2017 (10.1002/2017GL073057) who look at PDSI.

**See answer above.**

3. Figure 5 - given the inconsistencies in simulated vs. observed patterns particularly for moisture in North America, how much of the detection for moisture is being driven by the largely successful observed vs. simulation Mediterranean vs. northern European pattern?  The caption says that the moisture D&A refers to 'aggregate mean response grouped by the two regions of homogenous response indicated in Fig 4', but nothing is indicated (should there be a box or the region otherwise outlined?), and it isn't clear from the text alone (e.g. around Line 280) - given the mismatch in North America I note above and evident in Figure 4, I think the statement about detection and attribution in Line 310 and onward should probably be caveated - I suspect (and would ask the authors to establish if this is the case with some regional tests) the signal and successful Moisture D&A is being drive[n] by the Mediterranean/European pattern - the authors can also consult Fischer et al. 2007 and Rao et al. 2017.

**We agree that the results in Fig R3 are worth some further discussion and will consult the cited references. We will make further tests to study the weight of the European vs American pattern, and discuss the results in the revised manuscript. However, the simulated patterns in North America and in the Mediterranean are largely consistent in sign with observations, and perhaps this is clearer in confining the analysis to the most recent period for which there are more observations available (manuscript Fig 3; Fig R4).**

Minor comments:

Line 113: Just to verify: these are all tree-ring width data, and no density data correct, in Breitenmoser?

**Yes, correct.**

Line 119: suggest changing to 'As input to VSL we use the ...'

**We will change it to: "For the purpose of VSL parameter estimation, we use …"**

Line 159: suggest also citing the first paper on this, Cook, E. R., Briffa, K. R., Meko, D. M., Graybill, D. A., & Funkhouser, G. (1995). The 'segment length curse' in long tree-ring chronology development for palaeoclimatic studies. The Holocene, 5(2), 229-237.

Line 340: should probabl[y] add a citation near here to Stevenson, S., Fasullo, J. T., Otto-Bliesner, B. L., Tomas, R. A., & Gao, C. (2017). Role of eruption season in reconciling model and proxy responses to tropical volcanism. Proceedings of the National Academy of Sciences, 114(8), 1822-1826.

**Thanks for these suggestions. We will add these two references.**

Citation: https://doi.org/10.5194/cp-2021-80-RC2

---

## Author Response (AR2)

Dear Nerilie,

We have now addressed all technical corrections, which you have been asking for. You can see how we rephrased the start of the introduction and the new color scale of Figure 3 in the upload .pdf document including track changes.

Additionally, we realized during the preparation of the final version that very few sentences in the results and discussion sections can be written more clearly. Figure 6 now includes the previously lacking information on statistical significance. Finally, we uploaded all tree-ring simulations to NOAA's National Center for Environmental Information (NCEI) and included link to the data in the data availability section.

We hope the paper is now ready for publication.
Kind regards in the name of all coauthors,
Jörg